# Digitally virtualized atoms for acoustic metamaterials

Choonlae Cho[1,2,3], Xinhua Wen[1,3], Namkyoo Park[2] & Jensen Li[1]*

By designing tailor-made resonance modes with structured atoms, metamaterials allow us to obtain constitutive parameters outside their limited range from natural materials. Nonetheless, tuning the constitutive parameters depends on our ability to modify the physical structure or external circuits attached to the metamaterials, posing a fundamental challenge to the range of tunability in many real-time applications. Here, we propose the concept of virtualized metamaterials on their signal response function to escape the boundary inherent in the physical structure of metamaterials. By replacing the resonating physical structure with a designer mathematical convolution kernel with a fast digital signal processing circuit, we demonstrate a decoupled control of the effective bulk modulus and mass density of acoustic metamaterials on-demand through a software-defined frequency dispersion. Providing freely software-reconfigurable amplitude, center frequency, bandwidth of frequency dispersion, our approach adds an additional dimension to constructing non-reciprocal, non-Hermitian, and topological systems with time-varying capability as potential applications.

[1] Department of Physics, The Hong Kong University of Science and Technology, Clear Water Bay, Hong Kong, China. [2] Photonic Systems Laboratory, Department of Electrical and Computer Engineering, Seoul National University, Seoul 08826, Korea. [3] These authors contributed equally: Choonlae Cho, Xinhua Wen. *email: jensenli@ust.hk

O ver the past two decades, metamaterials have revolutionized how we manipulate classical waves, initially in the case of electromagnetic waves[1–3] and subsequently for acoustic waves[4–11], water waves[12] and, more recently, elastic waves in solids[13–17]. The ability of metamaterials to acquire physical properties beyond those of natural materials reflects the engineering degrees of freedom in designing artificial structures, e.g., split-rings, for resonance with tailor-made properties. Since then, many intriguing phenomena have been demonstrated, such as negative refraction and invisibility cloaking, which require the most extreme values of the constitutive parameters[18,19]. These effects confirm that metamaterials can be designed to yield a wide range of constitutive parameters and can be inhomogeneous. To make further use of metamaterials in practical situations, many applications (ranging from active invisibility cloaks to beam-scanning metamaterial antennas) require tunability or reconfigurability[20,21]. This can be achieved by optical pumping active materials, mechanically changing geometric parameters using MEMS, or combining external RLC circuit elements with meta-material structures. Tuning can also be extended to the level of each individual atom when backend electronics such as an FPGA chip or a computer are used to store and alter the state of the controlling parameters[22–24]. Nonetheless, for most of these metamaterials, tuning largely depends on the actual mechanism for modifying the metamaterial resonance of the physical structures. This poses a fundamental challenge in terms of the degree of flexibility or range of tunability, which is crucial in many applications requiring real-time reconfigurability. Additionally, it is hard to imagine using standard approaches for separately configuring resonating strength, bandwidth, and phase lag, e.g., for a Lorentzian frequency dispersion, as these depend on the actual tuning mechanisms.

In the context of tunability and reconfigurability, acoustic metamaterials mainly programmable by electronically controlled elements can be used to achieve a wide range of tunable effective parameters[25–29]. These acoustic metamaterials have proved useful as a platform for many intriguing wave phenomena, such as unidirectional invisibility[26], sound isolation[27], Willis coupling[28], and highly tunable mechanical properties[29]. These works point to the direction that programmable control with external circuits or microprocessors can be used to provide a higher level of abstraction of the physical properties of metamaterials.

Here, we introduce the concept of virtualization of metamaterials and demonstrate its application in manipulating acoustic wave propagation. By replacing the frequency resonating response of a physical metamaterial structure with a mathematically designed frequency dispersion implemented by digital convolution in the time domain within a microprocessing unit, the impulse response and the form of atomic response of a metamaterial structure is virtualized using a software code for digital representation. In the absence of any physical resonating structure, our digital representation of the virtualized metamaterial permits a highly arbitrary specification of the desired resonating frequency response. While the artificial physical structures of metamaterials mimic the working of natural atoms with engineering degrees of freedom, the concept of virtualization generalizes this analogy to a digital representation with tunability based solely on software modification, assigning another level of meaning to "meta".

## Results

**Virtualization of the signal response of meta-atom.** The virtualized acoustic metamaterial atom comprises a pair of circular microphones situated around two speakers (Fig. 1a) and is bonded on a small rectangular holder (see lower inset of Fig. 1a). This virtualized atom is then placed on the inner side of the top cover of the one-dimensional hollow waveguide to interact but without blocking the sound waves traveling within. For operation, the microphones and speakers are further connected to an external single-board computer (Raspberry Pi 3 with analog-to-digital/ digital-to-analog conversion module Waveshare ADS1256 and DAC8532). Sound waves arriving at the two microphones are detected, digitally sampled, and then processed in real-time by a software program running on the single-board computer. The resultant digital output signals are then converted back to analog and are feedback to the two speakers to generate the synthesized scattered waves. This combination of microphones, speakers, and the software defines the atom's generic scattering response.

Figure 1b shows a detailed representation of the software program. We construct a general linear operation from the signals at the two microphones $M_1$ and $M_2$ ($M_j(t)$) to the signals at the two speakers $S_1$ and $S_2$ ($S_i(t)$) as

$$S_i(t) = -\partial_t^2 \left( \tilde{Y}_{ij}(t - \delta t) * M_j(t) \right), \tag{1}$$

where * denotes convolution operator, and $\delta t$ represents an extra design time delay in the convolution operation. The whole operation comprises a matrix convolution and a differentiation in time to offset the result of convolution (kernel $\tilde{Y}$ to be designed later) as a driving voltage with zero averaged value for convenient handling within the program. A second time-derivative appears on this voltage since the speaker is actually driven by the voltage on a time-differential way. Finally, this time rate change of the voltage generates sound radiation by the speaker. In the frequency domain, the operation is summarized as

$$S_i(\omega) = Y_{ij}(\omega) M_j(\omega), \tag{2}$$

where $Y_{ij}(\omega) = \omega^2 \tilde{Y}_{ij}(\omega) e^{i\omega\delta t}$. Each orange arrow in the diagram connects a microphone to a speaker and is labeled as one of the matrix elements $Y_{ij}$ of the above operation (hereafter, "convolution"). The main horizontal line (in blue) represents the waveguide direction, in which an incident wave (e.g., from the left) travels and interacts with the atom. The secondary sources at $S_1$ and $S_2$ radiate symmetrically both forward and backward. These secondary radiations are added to the incident waves, finally becoming the reflected and transmitted waves within the waveguide. Having specified $Y_{ij}(\omega)$, it is possible to solve the overall response of the whole atom (Fig. 1b), yielding transmission/reflection coefficients and the polarizability matrix $\alpha_{ij}$ (or equivalently the scattering matrix $\mathcal{D}_{ij}$, see Supplementary Note 1) in terms of $Y_{ij}$. As the polarizability matrix in one-dimensional acoustics is generally $2 \times 2$, we chose to use two microphones and two speakers to detect and generate both monopolar and dipolar incoming and outgoing waves. Note that all digital computations when performing the convolution can apply only to a finite length of digital signal samples from $M_1$ and $M_2$ before the current digital signal sample and must finish within one sampling period (133 μs) of the analog-to-digital conversion module. We also note that while refs. [25–28] have setup the way to use electronic circuits to replace a physical structure, the further virtualization of the impulse response matrix in our case allows arbitrary specification of the atomic response (amplitude, center frequency, bandwidth, gain/loss, monopolar/dipolar type) and the frequency dispersion through program code without the need to set up different physical structures or different external circuits.

In modeling constitutive parameters (such as permittivity/ permeability in electromagnetism and mass density/ inverse bulk modulus in acoustics) for both natural materials and metamaterials, a Lorentzian frequency dispersion is probably the most representative spectral lines-shape. This acts like an "alphabet,"

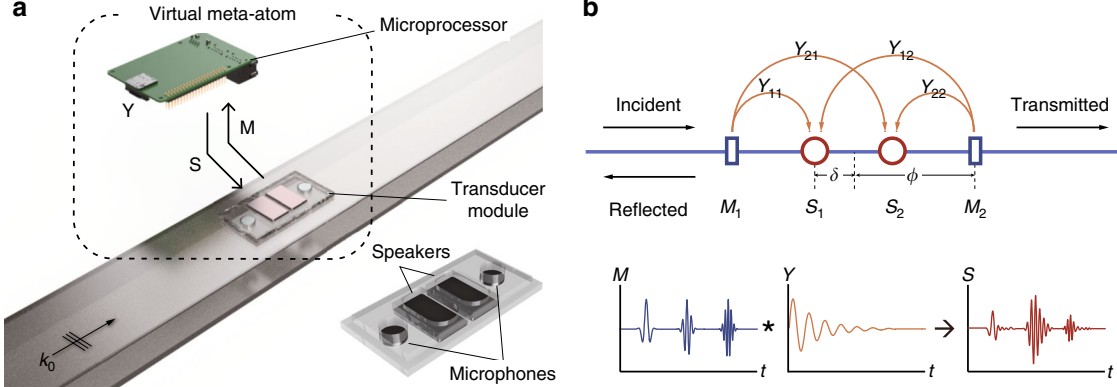

**Fig. 1 Schematics of virtualized metamaterial. a** Virtualized metamaterial consisting a structural atom of two circular microphones and two speakers (the two rectangular patches), connected to a small single-board computer for signal processing at a digital level. The virtualized metamaterial is embedded on the inner side of the top cover of a one-dimensional acoustic waveguide, not blocking the incident wave in a passive mode. **b** Schematic representation of the virtualized metamaterial atom: signals detected at the two microphones ($M_1$ and $M_2$) are convoluted with a $2 \times 2$ matrix ($Y$) resulting two signals to fire at the two speakers ($S_1$ and $S_2$) as secondary radiation from the atom. $Y$ is also called the impulse response of the atom. The phase distance between the two speakers is $2\delta$ (actual distance: $2 \times 8.5$ mm), and the phase distance between the two microphones is $2\phi$ (actual distance: $2 \times 2.6$ cm) for present implementation.

both for analytical modeling and as a numerical measure to decompose an arbitrary frequency spectrum to the sum of Lorentzian components of different spectral parameters. Here, we sought to instruct our virtualized metamaterial to mimic a Lorentzian response as our first example of a virtualized metamaterial. For simplicity, we focus on the monopolar response only, corresponding to an acoustic metamaterial with resonating bulk modulus; the relationship of the effective medium to atomic polarizability will be described later. A monopolar response of the virtualized atom is fulfilled by setting $\tilde{Y}_{11} = \tilde{Y}_{12} = \tilde{Y}_{21} = \tilde{Y}_{22} = \tilde{Y}/2$ in the software. We consider the convolution kernel $\tilde{Y}(t)$ to have the following form:

$$\tilde{Y}(t) = a\,\omega_0^{-2}\sin(\omega_0 t + \theta)e^{-\gamma t}\,(\text{for } t > 0) \text{ or } 0 \text{ (for } t \leq 0). \quad (3)$$

This involves several model parameters, where $\omega_0$ is resonating frequency, $\gamma$ is resonating bandwidth, and $a$ is resonance strength. In the formula, these have units of radial frequency and their values are specified in units of frequency by a factor of $1/2\pi$ for brevity. As an additional parameter to control the shape of frequency dispersion, we also defined $\theta$ as the "convolution phase." The software then connects the microphone signals to the speaker signals as in Eq. (2), generating the monopolar polarizability of the atom as

$$\alpha_{00} = \frac{2c}{i\omega}\mathcal{D}_{00}(\omega) = \frac{2c}{i\omega}\frac{2\cos\phi\cos\delta Y(\omega)}{1 - 2e^{i\phi}\cos\delta Y(\omega)} \cong \frac{4c}{i\omega}Y(\omega).$$

$$\text{with } Y(\omega) = \frac{\omega^2}{\omega_0^2}\frac{a}{2}\left(\frac{e^{i\theta}}{\omega_0 + \omega + i\gamma} + \frac{e^{-i\theta}}{\omega_0 - \omega - i\gamma}\right)e^{i\omega\delta t}, \quad (4)$$

where $c$ is the speed of sound in air, and $i$ is the unit imaginary number. All the other polarizability coefficients ($\alpha_{11}$, $\alpha_{01}$, and $\alpha_{10}$) should be zero in this case (see Supplementary Note 1). For a conventional metamaterial atom, we would expect the monopolar polarizability $\alpha_{00}$, or the inverse of bulk modulus to have a positive imaginary part for a passive atom, where $\theta = 0°$ and the convolution delay $\delta t$ is set to have $\arg(e^{i\omega\delta t}) \cong \pi/2$ at resonating frequency $\omega_0$, the resultant $\alpha_{00}$ mimics the Lorentzian frequency dispersion of a passive acoustic metamaterial in the frequency regime around $\omega_0$. As an example, choosing $\omega_0$ at 1 kHz, $\gamma$ at 15 Hz and a resonating strength $a$ at 7.85 Hz to implement a passive metamaterial ($\theta = 0°$), $\tilde{Y}(t)$ in Eq. (3) is then programmed as the convolution kernel in the virtualized metamaterial atom. To calculate the $\alpha$ or the $\mathcal{D}$ matrix, we measured the transmission

and reflection coefficients experimentally in both forward and backward directions within the waveguide. The blue curve in Fig. 2a represents the frequency trajectory from 750 to 1250 Hz of the experimentally extracted monopolar scattering coefficient $\mathcal{D}_{00}$. It traces what is roughly a circle, starting near the origin from small frequencies, in a counter-clockwise direction. It falls into the passive regime (indicated by the blue region), with the dashed circle passing through the origin with the center at $-0.5$. The complex transmission and reflection coefficients ($t$ and $r$) are simply related to $\mathcal{D}_{00}$ by $t - 1 = r = \mathcal{D}_{00}$. In this case, the resonance causes a dip in the transmission spectrum (blue curve in Fig. 2b). The Lorentzian shape of both its real and imaginary parts of the monopolar polarizability $\alpha_{00}$ is shown in Fig. 2c (blue curves and symbols). The symbols representing the experimental results agree well with the theoretical Lorentzian shape (lines: Eq. (4) with a slightly different value of $a$ fitted from experiment). This constitutes a conventional metamaterial that our virtualized metamaterial approach can mimic.

Although Eq. (3) is only a specific class of frequency dispersions, we can now change it by adopting other values of the convolution phase $\theta$ to obtain a distinctly different virtual metamaterial. Without needing to design a new physical structure as in the conventional approach to designing metamaterials, the software takes on the role of a physical structure. When $\theta$ is changed to 180°, $\tilde{Y}(\omega)$ simply flips signs. It produces an "anti-Lorentzian" shape of $\alpha_{00}$ (red curves and symbols in Fig. 2c). The imaginary peak then becomes negative, indicating a simulated material gain (see Supplementary Fig. 2 for the plot of power gain $|r|^2 + |t|^2$). More intuitively, in Fig. 2b, the transmission amplitude shows up as a peak beyond a value of one, with the additional power in the transmitted wave drawn directly from the external digital circuits. Figure 2a also shows the trajectory of $\mathcal{D}_{00}$ on the complex plane for the virtualized metamaterials at different convolution phases (e.g., $\theta = 90°$ and 270°); again, the trajectory is circular. In a geometric picture on the complex plane, the convolution phase $\theta$ actually rotates such circles about the origin by the same angle in a clockwise direction. This rotation on the complex plane moves part of the circular trajectory out of the passive zone, making the virtual atom unavoidably active. The virtualized metamaterial now takes on the original role of the swapped real and imaginary parts of the Lorentzian distribution. The real part of $\alpha_{00}$ shows up as a peak while the imaginary part shows up as an oscillation, respectively, shown as

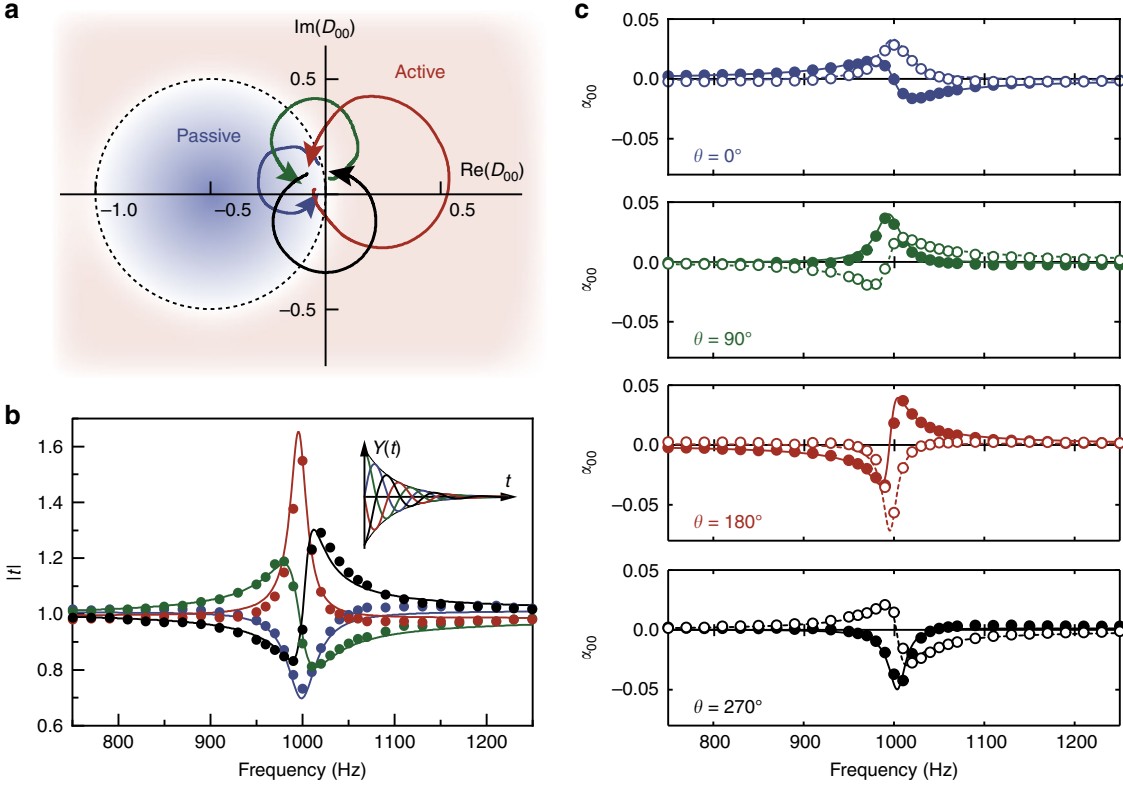

**Fig. 2 Mimicking Lorentzian frequency dispersion and active acoustic medium with a resonating monopolar response. a** Frequency trajectories of the experimental monopolar scattering coefficient $\mathcal{D}_{00}$ on the complex plane for 4 configurations with convolution phase $\theta = 0°$ (blue), 90° (green), 180° (red) and 270° (black). The red/blue shaded area denotes the active/passive region. Arrows indicate the direction from small to large frequencies. **b** Transmission amplitude spectrum for the four configurations. The convolution kernel $\tilde{Y}(t)$ for the four different cases of convolution phases are shown in the inset. **c** The real part (solid symbols) and the imaginary part (empty symbols) of the complex monopolar polarizability $\alpha_{00}$. Solid and dashed lines denote the corresponding theoretical Lorentzian line shapes for both the real and imaginary parts respectively.

green and black in Fig. 2c. For conventional metamaterials, a Fano resonance is usually introduced to provide an asymmetric line-shape[30]. Here, we can create an asymmetric line-shape (see $\theta = 90°$ and 270° in the $|t|$ spectrum) by tuning the convolution phase value.

The virtualized representation of the metamaterial Eq. (3) provides a straight-forward implementation of an active medium. One interesting point is that the anti-Lorentzian shape (effectively the same as a Lorentzian shape but with a negative resonating strength $a$) has to stand as an approximation in the frequency regime around the resonating frequency. If valid for the whole frequency axis, the poles of the complex function $Y(\omega)$ will occur entirely in the upper half complex plane, denying causal implementation of the convolution kernel. Our approach guarantees causality because it implements the virtual atom by convolution in the time domain. The approximation of the anti-Lorentzian shape around the resonating frequency is linked to the condition $\arg(e^{i\omega\delta t}) \cong \pi/2$ which is only approximately satisfied.

**On-demand tuning of dispersion**. Our virtualized approach to constructing metamaterial allows us to freely reconfigure the frequency dispersion on-demand in a very flexible way. Conventionally, a physical metamaterial design provides both resonating strength and bandwidth at the same time. In principle, these two physical properties (or model parameters) can be reconfigured by two geometric parameters of the metamaterial. However, decoupled control of the two physical properties by two geometric parameters is highly non-trivial[31]. While varying a single geometric parameter often results in a simultaneous change

in both physical properties, our approach means that resonating strength and bandwidth are simply two input parameters that can be specified independently, as the convolution kernel ($\tilde{Y}(t)$ in Eq. (3)) is defined simply as a mathematical function in the software code for digital representation. Figure 3a shows the virtualized metamaterial as specified schematically in Fig. 2. The resonating strength $a$ is varied from 3.93, 7.85 to 11.78 Hz while the resonating bandwidth is fixed at $\gamma = 15$ Hz. The magnitude and spectral profile of both the real part (solid lines and filled symbols) and the imaginary part (dashed lines and empty symbols) of the monopolar polarizability increase and scale with $a$. Similarly, we reduced the resonating bandwidth $\gamma$ from 30 to 15 and 7.5 Hz to obtain sharper resonance with $a$ being kept at a constant value of 7.85 Hz. The results are shown in Fig. 3b; in both cases, the experimentally obtained frequency dispersions of monopolar polarizability $\alpha_{00}$ (plotted in symbols) agree well with the theoretical model derived from $Y(t)$, where the solid and dashed lines represent its real and imaginary parts, respectively. In fact, as the magnitude of $Y(t)$ decays in time through $\exp(-\gamma t)$, the smallest $\gamma$ we can achieve is limited by the total convolution time ($T_c$) implemented in the software code. A smaller $\gamma$ requires a larger $T_c$ if the magnitude of $Y(t)$ is to decay to a negligible value before truncation. For example, a requested 10 dB decay in $Y(t)$ before truncation was chosen for accurate implementation of the target $Y(t)$ with $\gamma$ as small as 3.4 Hz. In these cases, the resonating frequency was kept at 1 kHz. Finally, we fixed $\gamma = 15$ Hz and $a = 7.85$ Hz and then varied the resonating frequency $\omega_0$ from 0.8 to 1.2 kHz in steps of 100 Hz. Clear resonances were observed around the designated resonating frequencies, with a tunable range of resonating frequencies approaching almost 40% of the

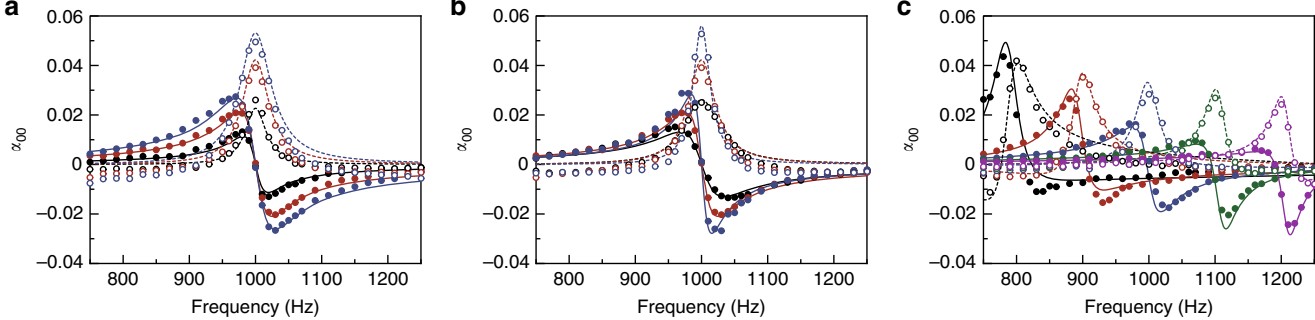

**Fig. 3 Decoupled tuning of resonance amplitude, bandwidth, and center frequency for virtualized metamaterial. a** Three cases of resonating strength $a = 3.93$ (black), $7.85$ (red) and $11.78$ Hz (blue) with constant resonating bandwidth $\gamma = 15$ Hz and resonating frequency $\omega_0 = 1$ kHz. The real/imaginary part of monopolar polarizability $\alpha_{00}$ is plotted in solid/empty symbols for the experimental results. **b** Three cases of resonating bandwidth $\gamma = 30$ (black), $15$ (red) and $7.5$ Hz (blue) with constant resonating strength $a = 7.85$ Hz. **c** Resonating frequency $\omega_0$ varies from $800$ to $1200$ Hz in steps of $100$ Hz with $\gamma = 15$ Hz and $a = 7.85$ Hz. Here an extra phase shift is inevitable because of the inherent time delay in electronic devices as the resonance frequency increases. The corresponding theoretical models are plotted in solid/dashed lines for the real/imaginary part in all panels.

central frequency in the tunable range ($\Delta\omega/\omega$), which is limited only by the speed of the electronics. Faster electronics can further increase the digital sampling frequency to achieve a higher frequency bound while the convolution (accomplished digitally within one sampling period) can involve more samples. We also note that the tunability offered by our approach can become more flexible and generic. As $Y(t)$ is a mathematical function freely encoded in the software, we can render the frequency dispersion to have more general shape, for example, to capture multi-resonating frequencies, each with different strengths, bandwidths and with either gain or loss.

**Independent control of monopolar and dipolar scattering.** Connecting monopolar incidence to monopolar scattered waves corresponds to an acoustic metamaterial with a resonating bulk modulus[5,6]. Our virtualized approach can also be used to construct metamaterials with a more general response than monopolar scattering. As our atom has sufficient degrees of freedom when generating both monopolar and dipolar secondary radiations, the same virtualized metamaterial technique can be used to generate a dipolar scattering response, corresponding to an effective resonating density. In this case, we set $\tilde{Y}_{11} = -\tilde{Y}_{12} = -\tilde{Y}_{21} = \tilde{Y}_{22} = \tilde{Y}/2$ and the dipolar scattering coefficient is then given by

$$\alpha_{11} = \frac{2c}{i\omega}\mathcal{D}_{11}(\omega) = \frac{2c}{i\omega}\frac{2\sin\phi\sin\delta Y(\omega)}{1 + 2ie^{i\phi}\sin\delta Y(\omega)} \cong \frac{4c}{i\omega}\sin\phi\sin\delta Y(\omega).$$

$$(5)$$

$\tilde{Y}(t)$ and $Y(\omega)$ are still defined in Eq. (3) and in Eq. (4) (with subscript 1 added to $a$ and $\gamma$ to indicate the dipolar nature of the model parameters). To demonstrate, we set a resonating frequency $\omega_0 = 1.2$ kHz, resonating strength $a_1 = 14.25$ Hz and linewidth $\gamma_1 = 8$ Hz; we also set the convolution phase $\theta = 0°$, corresponding to the passive case. The resultant real and imaginary parts of $\alpha_{11}$ are shown in Fig. 4a as the black solid and dashed curves with resonating behavior. This corresponds to a resonating mass density (in an effective medium of the virtualized metamaterial) with a positive resonating peak in its imaginary part. On the other hand, if we change the convolution phase $\theta$ to $180°$ (with the same parameters for $\omega_0$, $a_1$ and $\gamma_1$), the resonating atoms are gain-dominating around the resonating frequency, showing a negative peak in the imaginary part of $\alpha_{11}$ in Fig. 4b. In the same Fig. 4a, b, the corresponding values of monopolar response $\alpha_{00}$ (shown in red) have much smaller amplitudes than the instructed dipolar response.

By exploiting the virtualized metamaterial's degrees of freedom, the monopolar resonance and dipolar resonance can be generated at the same time. More importantly, all of the resonating model parameters can be designed as highly arbitrary. For the implementation, we set $\tilde{Y}_{11} = \tilde{Y}_{22} = (\tilde{Y}_0 + \tilde{Y}_1)/2$ and $\tilde{Y}_{12} = \tilde{Y}_{21} = (\tilde{Y}_0 - \tilde{Y}_1)/2$, where $\tilde{Y}_0$ and $\tilde{Y}_1$ are implemented by Eq. (3) with resonating strength $a_0$ and $a_1$ and resonating linewidth $\gamma_0$ and $\gamma_1$, with the resonating frequency commonly set at $\omega_0 = 1.2$ kHz. As shown in Fig. 4c, both $\mathcal{D}_{00}$ and $\mathcal{D}_{11}$ are now resonating. Model values are detailed in the caption to Fig. 4. The virtualized atom can also be immediately transferred to the gain regime by changing $\theta$ from $0°$ to $180°$, as shown in Fig. 4d; the resonating peak of the imaginary part for both $\alpha_{00}$ and $\alpha_{11}$ goes negative as a dominating gain around resonance. We have been using the polarizability to represent the atomic property. On the other hand, our 1D metamaterial can be equivalently represented as an effective medium of thickness $d$ (actual thickness of our atom = $6.5$ cm), the relationship between the effective bulk modulus $B$ and the effective mass density $\rho$ can be related to the monopolar and dipolar polarizabilities as

$$\chi_0 = B_0/B - 1 \cong \alpha_{00}/d,$$
$$\chi_1 = \rho/\rho_0 - 1 \cong \alpha_{11}/d,$$

$$(6)$$

where $B_0$, $\rho_0$, are the bulk modulus, mass density of the air, respectively, while $\chi_0$ and $\chi_1$ refer to monopolar and dipolar susceptibility (see Supplementary Fig. 3). The ability to control both monopolar and dipolar polarizabilities is essential in order to control simultaneously the transmission and reflection amplitudes through $t - 1 = \mathcal{D}_{00} + \mathcal{D}_{11}$ and $r = \mathcal{D}_{00} - \mathcal{D}_{11}$. We note that the near-field coupling (as there is no physical structure), if we periodically place identical atoms along the propagation direction, can be neglected, the effective medium parameters are still valid when we scale up the number of atoms (see Supplementary Fig. 6 and the results about multiple unit cells in Supplementary Note 3).

Unlike conventional metamaterials that require the design of a special kind of atom, the virtual implementation of metamaterials allows density and modulus to be independently tuned without affecting each other and without modifying any physical structures or external circuits. Figure 4e, f show the corresponding results for the model parameters in Fig. 4c, d ($\theta = 0°$ and $180°$) but with the resonating strength $a_0$ divided by a factor of $1.6$. The results show that dipolar resonance is almost unaffected while monopolar resonance (e.g., the peak of Im($\alpha_{00}$)) is divided by roughly the same factor. Our results confirm the advantages

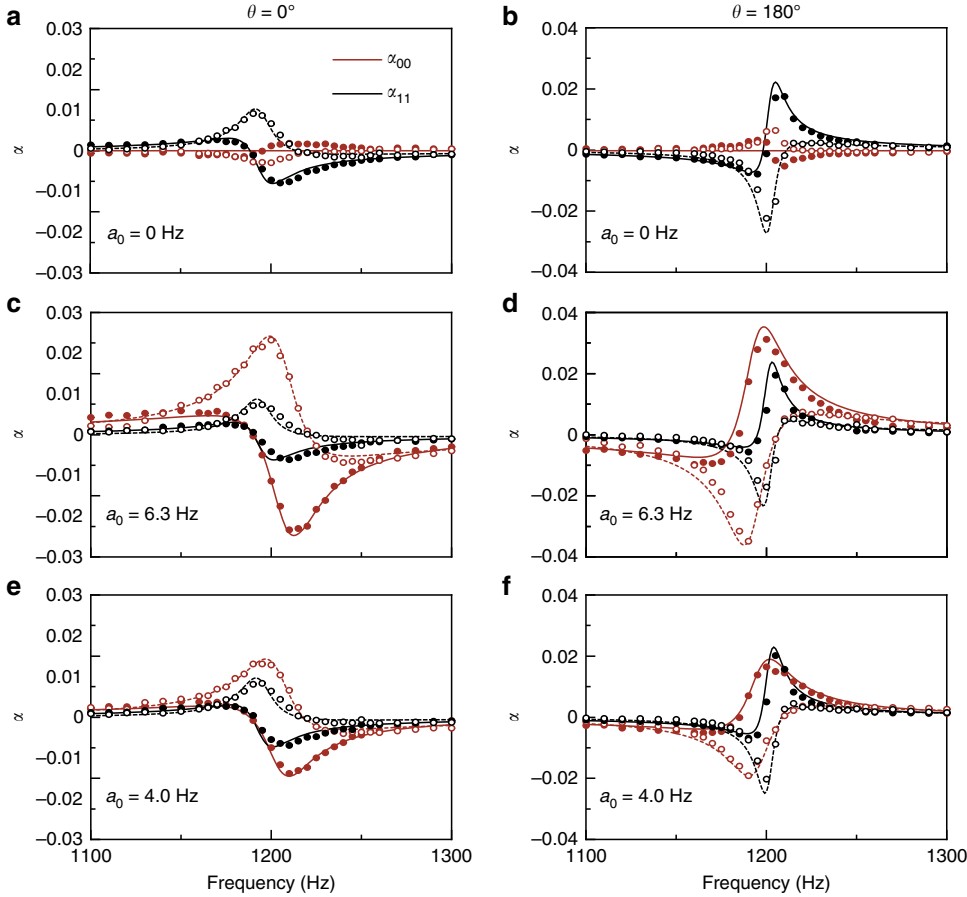

**Fig. 4 Decoupled control on the monopolar and dipolar scattering coefficients. a, b** Virtualized metamaterial with only dipolar response where the model parameters are set as $\gamma_1 = 8$ Hz, $a_1 = 14.2$ Hz, resonating frequency $\omega_0 = 1.2$ kHz and convolution phase $\theta = 0°$ (**a**) and 180° (**b**). Polarizabilities $\alpha_{11}$ and $\alpha_{00}$ are plotted in black and red colors. Solid and dashed lines denote the real and imaginary parts of the theoretical line shapes. Solid and empty symbols denote the real and imaginary parts of the experimental results, respectively. Resonating dipolar $\alpha_{11}$ dominates over monopolar $\alpha_{00}$. **c, d** Monopolar response is further added to configurations in (**a, b**) with model parameters $\gamma_0 = 15$ Hz, $a_0 = 6.3$ Hz with the same $\omega_0$. **e, f** Monopolar response is changed to $a_0 = 4$ Hz while other model parameters are kept the same. For all results, the left/right panel shows the scattering coefficients for convolution phase $\theta = 0°$ (**a, c, e**) and 180° (**b, d, f**).

of the virtualization approach in designing tailor-made config-urations, addressing some of the inherent limitations of conventional metamaterial approaches by allowing the model parameters to be tuned to any desired value. This also contrasts with common approaches in which the resonating strength and bandwidth of active metamaterials are unlikely to be indepen-dently configurable because they depend on actual mechanisms to achieve gain. Moreover, while dipolar resonance is much sharper than monopolar resonance in conventional metamaterials, the virtualized approach can make the two resonances are similar in shape and bandwidth (see Fig. 4e). This enables impedance matching (to achieve small reflectance) in a wide frequency regime (Supplementary Fig. 4). In short, the present virtualized approach offers great advantages for modifying metamaterial resonance.

## Discussion
In conclusion, we have proposed and provided experimental support for the concept of virtualized metamaterials, removing the physical restrictions of traditional metamaterials. Using a convolution kernel function and digitally driven wave sources to directly synthesize the scattered wave, it was possible to freely access different frequency dispersion curves on demand, achiev-ing decoupled control on different wave parameters and

constitutive parameters. The software-controlled transition between Lorentzian, anti-Lorentzian, and asymmetric dispersion curves were experimentally confirmed within a single platform, while independently addressing amplitude, center frequency, bandwidth, and convolution phase for all dispersion curves across a broad frequency range. The frequency dispersion, equivalently the impulse response function, can be programmed to other shapes to achieve optimal bandwidth for material constitutive parameters[25], material gain, zero index, etc. In fact, the frequency dispersion of the material parameters can be further modulated in time slowly (comparing to the sampling period), we can then apply such dynamic modulation on individual atoms to construct time-varying metamaterials[32,33]. For example, a modulation phase lag between different atoms can be used to generate an Aharonov–Bohm phase and non-reciprocal transmission[34,35], which can now be readily achieved in acoustics. Furthermore, we can also use an ensemble of these virtualized atoms to realize Floquet topological phases with a temporally periodic Hamilto-nian[36,37]. For our platform, gain and loss can also be matched exactly and varied in time domain due to the flexible tunability, allowing us to investigate non-Hermitian systems[38,39] with exceptional points that can now be scanned through dynamically and without any physics structures. It should also be straight-forward to inverse-derive the mathematical kernel for a virtua-lized metamaterial on demand, for targeted applications and wave

parameters. This approach is not limited to the acoustic platform, as the implementation of the convolution kernel function can also be envisaged in FPGA, for the faster convolution required in ultrasonic or microwave applications.

## Methods

**Measurement setup and metamaterial structure.** The experimental set-up is schematically shown in Fig. 1a. A meta-atom consists of two speakers and two microphones with electric peripherals including a microprocessor, analog-to-digital/digital-to-analog converter and amplifying modules. For the digital convolution, the microprocessor is programmed to operate at a sampling frequency of 7.5 kHz and using 400 sampling data to accomplish all calculation process within one cycle. Speakers and microphones which are connected to the microprocessor and communicate through SPI (Serial Peripheral Interface), are assembled in an acrylic frame (width = 3 cm, length = 6.5 cm). This transducer module is mounted on top of a one-dimensional rectangular acoustic waveguide (width = 6 cm, height = 2 cm). For the measurement, we used the 4-points measurement method with the National Instrument DAQ device and Labview system. To obtain all the scattering parameters, the acoustic wave incident from one end of the waveguide, is measured at each frequency for each of two different boundary conditions. The scattering parameters spectra can also be obtained by doing a transient stimulus using a wave packet of finite duration and a carrier frequency (see Supplementary Note 4).

## Data availability
The data that support the plots within this paper and other findings of this study are available from the corresponding author upon reasonable request.

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

## Acknowledgements
J.L. acknowledges funding from the Research Grants Council under Grant No. 16303019. N.P. was supported by the NRF of Korea through the Global Frontier Program (2014M3A6B3063708).

## Author contributions
J.L. conceived the idea of virtualization of metamaterial and initial design. C.C., J.L., and X.W. establish the setup of atom and program control. C.C. and X.W. made the measurements. All authors contributed in data analysis and writing the manuscript. J.L. and N.P. manage the project.

## Competing interests
The authors declare no competing interests.
