## [Peer Review File · Nature Communications]

Reviewers' Comments:

Reviewer #1:

Remarks to the Author:

The manuscript presents the design of a metamaterial unit cell whose acoustic behavior is set by microcontroller-driven electronics as opposed to geometrical features. The manuscript shows that this approach is an alternative to the passive monopole and dipole resonators on which most acoustic metamaterials are based. The presentation is mostly clear, very well-written, and the topic is of interest to a large audience. In principle, I would be in favor of its publication in Nature Communications. However, there are several issues that should be addressed before that.

- 1) There are other published papers that report metamaterials whose acoustic properties are programmable via electronics and microcontrollers. In particular Refs. 27 and 28 report metamaterials made of unit cells that appear to be very similar to what is presented here. The authors should clearly state the differences between their cell and what has been done in the past.
- 2) The goal of this work is to show that the basic Lorentzian resonances occurring in most passive metamaterials can be replicated using programmable active unit cells. However, the resonant behavior is often an undesirable property needed by passive metamaterials to achieve strong acoustic responses. The appeal of active metamaterials is that they do not require resonances and can (in principle) be broadband. Could the unit cell presented here be used to implement monopolar and dipolar non-resonant responses?
- 3) The sentence "Nonetheless, for all such metamaterials, tuning largely depends on the actual mechanism for modifying the metamaterial resonance of the physical structure" is incorrect and should be corrected. For example, Refs. 25-28 do not alter the physical structure, and [27,28] do not even present resonant metamaterials.
- 4) What is the physical meaning of the second time derivative in eq 1?
- 5) The gain of a material cannot be judged by looking only at the reflection coefficient (see page 7, line 143). The manuscript (or the supplementary document) should show $|r|^2 + |t|^2$ to determine whether the unit cell introduces gain.
- 6) What is the physical meaning of the variables a_0 , a_1 , b_0 , and b_1 in the supplementary document? As far as I can tell they are never introduced in the text.

Reviewer #2:

Remarks to the Author:

Summary: The primary contribution of this paper is the concept of a virtual meta-atom whose bulk modulus and mass behavior is programmable via software. The authors then create and experimentally-verify that they can achieve Lorentzian and anti-Lorentzian frequency dispersion for a single meta-atom.

Specific Review Comments:

1. The paper may not be as novel as the authors believe. The Hopkins group at UCLA has published multiple articles on "robotic" metamaterials that report similar functionality - i.e., the metamaterial is programmable via software. The following references are worth consulting:
 - a) Programmable Elastic Metamaterials, Advanced Engineering Materials, 2015
 - b) Design and fabrication of a three-dimensional meso-sized robotic metamaterial with actively controlled properties, Mater. Horiz., 2019
 - c) others

2. Although the resonant frequency of the meta-atoms is given, the response rate (which is different) is not detailed. Can the meta-atom also respond to a transient stimulus in a time faster than this? This would be necessary to truly replicate metamaterial behavior.

3. Metamaterials have their own unique language, almost exclusively presented in the form of band structure. The authors pursue an entirely different presentation (in terms of r/i curves). It isn't clear how to translate these to band structure, and furthermore, it isn't clear that the single meta-atom presentation suggests much about a metamaterial, which would require many repeated unit cells of meta-atoms. For these two reasons, the paper is highly disconnected from its stated topic, in both presentation and application. The authors are encouraged to seek a more traditional presentation of the results, and to expand their work to multiple unit cells.

Reviewer #3:

Remarks to the Author:

This manuscript is concerned with the tunability of metamaterials in regard to the characteristic propagation of elastic waves. The authors identify several tuning methods from the literature (e.g., active material constituents) and argue that the time delay and range of response are hardly suitable for applications which require real-time adjustments. The authors propose a digitally constructed meta-atom -- consisting of a microphone-speaker assembly attached to a 1D waveguide -- for independently tuning the effective mass density and bulk modulus critical to the wave dynamics. Signals registered by the microphones are convoluted (in time) via a defined transformation and sent to the speakers as secondary monopole (linked to effective modulus) or dipole (linked to effective mass density) radiation. The theoretical/numerical results were experimentally confirmed. The manuscript is clearly written, and presents original and potentially transformative concepts. Consequently, the review recommends the manuscript for publication.

Small edit suggestion: Equation 4 displays the imaginary unit "i" in both scripted and non-scripted form. One form should be used.

Thanks for all the reviewers for the very constructive comments. We have now addressed them in the following point-by-point. The modification in text are highlighted by blue color.

Reviewer #1 (Remarks to the Author):

*The manuscript presents the design of a metamaterial unit cell whose acoustic behavior is set by microcontroller-driven electronics as opposed to geometrical features. The manuscript shows that this approach is an alternative to the passive monopole and dipole resonators on which most acoustic metamaterials are based. **The presentation is mostly clear, very well-written, and the topic is of interest to a large audience. In principle, I would be in favor of its publication in Nature Communications.** However, there are several issues that should be addressed before that.*

Reply: Thanks for the positive comment to our work. Below we will address the question about the advance of our approach and the detailed comments:

*1) There are other published papers that report metamaterials whose acoustic properties are programmable via electronics and microcontrollers. In particular **Refs. 27 and 28 report metamaterials made of unit cells that appear to be very similar to what is presented here.** The authors should clearly state the differences between their cell and what has been done in the past.*

Reply: As the reviewer points out, Refs. 27 and 28 belong to this category of works using electronics or microcontrollers to control acoustic metamaterials. In all these cases, the feedback is an external amplifying circuit (e.g. an opamp, to define the impulse response). A microcontroller is to control the system parameters, such as a programmable cut-off frequency of a filter (to the amplifying circuit), a programmable constant-phase delay, or a switching capability while the response function is not arbitrary.

We regard these are the classics, inspiring further virtualization of “all” properties of a metamaterial, including frequency dispersion and the form of various atomic responses (amplitude, center frequency, bandwidth, gain/loss). **In our work, by specifying a designer / generic impulse response matrix that with a program code (not confined by the form of external amplifying feedback circuit), we virtualize the signal response specification.** Therefore, we do not need to choose different physical constructions to change resonating response (the impulse response is arbitrary here) or to change from density to modulus response (the form of matrix to define atomic response is arbitrary), etc. **We also note that we use non-blocking transparency mode here: transmission in off-state.** In this way, the secondary radiation generated from digital feedback are intuitively mimicking the scattering response of conventional metamaterial atoms. This is the key for us to formulate a feasible scheme of such virtualization.

Now, we have added the following discussion below Eq. 2 after introducing our scheme of virtualization:

“We also note that while Refs 25-28 have setup the way to use electronic circuits to replace a physical structure, the further virtualization of the impulse response matrix in our case allows arbitrary specification of the atomic response (amplitude, center frequency, bandwidth, gain/loss, monopolar/dipolar type) and the frequency dispersion through program code without the need to set up different physical structures or different external circuits.”

2) *The goal of this work is to show that the basic Lorentzian resonances occurring in most passive metamaterials can be replicated using programmable active unit cells. However, the resonant behavior is often an undesirable property needed by passive metamaterials to achieve strong acoustic responses. The appeal of active metamaterials is that they do not require resonances and can (in principle) be broadband. Could the unit cell presented here be used to implement monopolar and dipolar non-resonant responses?*

Reply: We agree that the freedom of active metamaterial in defining impulse response (can be resonating or not) can benefit bandwidth discussion for other applications. Here, we give an example about broadband near zero refractive index (NZI) by defining an impulse response other than Lorentzian: for this, we assign $Re(\rho) \cong 0$ over target frequency range (f_1, f_2) and derive their imaginary part from Kramers-Kronig relations (following causality condition). The time domain response of this designer NZI dispersion metamaterial becomes then $\frac{1}{t} [\sin(2\pi f_2 t) - \sin(2\pi f_1 t)]$, different from the Lorentzian resonance. **The following figure demonstrates the experimental results that the flat mass density can cover a broad range of frequencies, rather than a narrow band phenomenon, proving the flexibility of our dispersion control, from the top-down approach.**

We feel that the above application, a bit far from the current demonstration of resonating metamaterials, still needs further elaboration to become a new work (we are currently working on it, and we sincerely thank the reviewer for the suggestion). But we have added some discussion in conclusion to give some insight for future direction:

“The software-controlled transition between Lorentzian, anti-Lorentzian, and asymmetric dispersion curves were experimentally confirmed The frequency dispersion, equivalently the impulse response function, can be programmed to other shapes to achieve optimal bandwidth for material constitutive parameters²⁵, material gain, zero index, etc.”

3) *The sentence "Nonetheless, for all such metamaterials, tuning largely depends on the actual mechanism for modifying the metamaterial resonance of the physical structure" is incorrect and should be corrected. For example, Refs. 25-28 do not alter the physical structure, and [27,28] do not even present resonant metamaterials.*

Reply: **We agree that we were not careful in making such a statement. The sentence should be only about two different types of tunable metamaterials (Refs. 22-24): 1. tunable resonating physical structures or 2. coding metamaterials that a computer/FPGA to select atomic configurations from a discrete set of geometries of physical structures. We have now moved the sentence to the end of first paragraph about Refs. 22-24. On the other hand, we have now introduced Refs. 25-28 along the route of programmable metamaterials with**

programmable control either through external circuits/microprocessor. These inspire our virtualization of the impulse response (frequency dispersion) in this work:

“In the context of tunability and reconfigurability, acoustic metamaterials mainly programmable by electronically controlled elements can be used to achieve a wide range of tunable effective parameters [25-29]. ... These works point to the direction that programmable control with external circuits or microprocessors can be used to provide a higher level of abstraction of the physical properties of metamaterials.”

4) *What is the physical meaning of the second time derivative in eq 1?*

Reply: One of the time derivatives is to eliminate the DC component, making it easier to use the full dynamic range of the voltage in driving the speaker (mentioned in main text). Another time-derivative appearing on the same Eq. 1 is related to the manner of the speaker being driven by the voltage in a time-differential way (a constant voltage does not generate sound). Total of two time-derivatives may not be necessary. It is rather for our convenience of operation. We have now added the following in explaining the second time derivative:

“A second time-derivative appears on this voltage since the speaker is actually driven by the voltage on a time-differential way.”

5) *The gain of a material cannot be judged by looking only at the reflection coefficient (see page 7, line 143). The manuscript (or the supplementary document) should show $|r|^2 + |t|^2$ to determine whether the unit cell introduces gain.*

Reply: Thank you for your kind suggestion. We have now added Fig. S2 in supplementary information for the plot of $|r|^2 + |t|^2$ for the configurations in Fig. 2 and refer the readers to Fig. S2 in main text. It is well above value one to indicate gain now:

Figure S2 | Metamaterial power gain The spectrum for the sum of transmission intensity and reflection intensity for the 4 configurations (in Fig. 2) with convolution phase $\theta = 0^\circ$ (blue), 90° (green), 180° (red) and 270° (black), where symbols and lines denote the experimental and theoretical results respectively. Here $|r|^2 + |t|^2 - 1$ is the power gain for one-side incidence.

6) *What is the physical meaning of the variables a_0 , a_1 , b_0 , and b_1 in the supplementary document? As far as I can tell they are never introduced in the text.*

Reply: Thank you for your careful reading of our manuscript. We have now added discussion of these symbols in Supplementary section A to explain the various symbols in Eq. S1:

“The monopolar (dipolar) incident waves are denoted by a_0 (a_1) while the monopolar (dipolar) scattered waves are denoted by b_0 (b_1) in generating symmetric (antisymmetric) waves, see Fig. S1 for the schematic representation.”

Reviewer #2 (Remarks to the Author):

Summary: The primary contribution of this paper is the concept of a virtual meta-atom whose bulk modulus and mass behavior is programmable via software. The authors then create and experimentally-verify that they can achieve Lorentzian and anti-Lorentzian frequency dispersion for a single meta-atom.

Reply: Thanks for the questions for seeking further understanding. We hope the below can clarify and provide some satisfactory answers.

Specific Review Comments:

1. The paper may not be as novel as the authors believe. The Hopkins group at UCLA has published multiple articles on "robotic" metamaterials that report similar functionality - i.e., the metamaterial is programmable via software. The following references are worth consulting:

- a) Programmable Elastic Metamaterials, Advanced Engineering Materials, 2015*
- b) Design and fabrication of a three-dimensional meso-sized robotic metamaterial with actively controlled properties, Mater. Horiz., 2019*
- c) others*

Reply: Thanks. Reviewer 1 also points out there are works using microprocessors for programming metamaterials. **We have now cited Hopkin's Mater. Horiz. paper along Refs. 25-28 on the foundation on using external / microprocessor circuits inspiring us on the new level of virtualization of impulse response matrix of the metamaterials.** We hope it helps in clarifying the position of our paper. We have added in introduction:

"In the context of tunability and reconfigurability, acoustic metamaterials mainly programmable by electronically controlled elements can be used to achieve a wide range of tunable effective parameters [25-29]. ... These works point to the direction that programmable control with external circuits or microprocessors can be used to provide a higher level of abstraction of the physical properties of metamaterials."

Another approach from Hopkin's group is to use a network of beams to assist buckling in switching between different phases of mechanical properties [Nature Commun., 9, 4594 (2018)]. Further programmable control is added if each beam in the network can be switchable between two states using electromagnets [Programmable elastic metamaterials. Advanced Engineering Materials, 18, 643 (2016)]. This work constitutes to the category about coding metamaterial (Refs. 23,24) where individual state of the atomic structure can be stored digitally and controlled by a microprocessor/FPGA. The paper is now also cited along Refs. 23,24.

Our focus is on the dynamic response and the feedback between microphones and speakers are in contrast to Hopkin's works that employ actuators in *deforming the shape or structure for tuning static stiffness*. Sensors in his case are for monitoring the displacement/strain without feedback.

2. Although the resonant frequency of the meta-atoms is given, the response rate (which is different) is not detailed. Can the meta-atom also respond to a transient stimulus in a time faster than this? This would be necessary to truly replicate metamaterial behavior.

Reply: Thank you very much for your valuable question. We confirm that our metamaterial can respond to transient stimuli and it can respond faster than the response time of the resonating metamaterial.

Figure S7 | Response for step-type input at fixed frequency Typical response time to approach steady state with amplitude response agreeing to target spectrum for (a) resonating linewidth $\gamma = 7.5$, (b) 15.0, and (c) 30.0Hz for same resonating frequency 1kHz, and the same incident step function with carrier frequency 1kHz in experiments. Smaller γ has a smaller response time. The time constant for each bandwidth is given by 29ms, 14ms, and 7ms.

From Fig. S7, a steady state of expected amplitude can be achieved after some time from the beginning of a step function input (multiplied by a carrier frequency at 1kHz). Smaller resonance linewidth (high Q resonator) requires longer time to respond. For the case of resonating metamaterial used in Fig. 2 with $\gamma = 15$ Hz (Fig. S7b), the response time is experimentally found as 14ms. **We need a response time proportional to $1/\gamma$ for our resonating metamaterial.**

Figure S8 | Transient response for finite pulses (a) The input (quadratic spline) pulses with finite duration 5ms or 2.5ms, and carrier frequency 1kHz as excitation (incident waves). (b) and (c) The monopolar polarizability measured from the metamaterial in Fig. 2 with convolution phase $\theta = 0^\circ$ and resonating linewidth $\gamma = 15.0$ Hz.

Next, we apply a transient excitation. Here, the responses from 750 to 1250Hz are measured and obtained in one single transient experiment by inverse Fourier transforming the measured signal at various microphones. Figure S8(a) shows the input pulse (quadratic spline) with duration 5ms and 2.5ms. The carrier frequency of the pulse is set as 1kHz. Such an incident transient pulse is fired to the metamaterial, with configuration in Fig. 2 with convolution phase $\theta = 0^\circ$ and resonating linewidth $\gamma = 15$ Hz. Figures S8(b) and S8(c) show the corresponding experimental spectrum (solid/empty symbols for real/imaginary part) obtained for the monopolar polarizability. As we can see, the results follow the theoretical spectrum (in lines), even up to a pulse width as short as 5ms, which is much shorter than the response time 14ms,

indicating the metamaterial can work for transient excitations. When the pulse width is further reduced, we see higher noise as the total power of the incident wave is now spread across a wider range of frequencies.

For the non-resonating case, there will still be a response time, not related to the resonance but to our digital implementation. Ultimately, due to **Shannon sampling theorem: in order to read and respond to the input frequency in the Shannon theoretical limit, we will need at least 2 x sampling period T_s** . However, it would require 4~8 sampling points for a cycle. We measured the scattered wave response from the atom, for the input of short pulse (pulse width $\sim 4\text{ms}$, $f \sim 1\text{kHz}$). Figure S9 shows there exist time delay ($500\mu\text{s} \sim 4T_s$) between the atom-off state and atom-on state, when measured at the receiver at the end of the waveguide, which includes both physical and electronic delay times.

Figure S9 | Response time for a non-resonating metamaterial with small but broadband monopolar scattering coefficient. Incident (p_i) and scattering (p_s) waves of a non-resonating metamaterial in probing the ultimate response time, due to all physical and digital electronic delay.

We have now put the above discussion and figures in a new section “Transient response of the metamaterial” in Supplementary Information and refer to it in main text.

*3. Metamaterials have their own unique language, almost exclusively presented in the form of band structure. The authors pursue an entirely different presentation (in terms of r/i curves. It isn't clear how to translate these to band structure, and furthermore, it isn't clear that the single meta-atom presentation suggests much about a metamaterial, which would require many repeated unit cells of meta-atoms. For these two reasons, the paper is highly disconnected from its stated topic, in both presentation and application. **The authors are encouraged to seek a more traditional presentation of the results, and to expand their work to multiple unit cells.***

Reply: We apologize our use of many nonconventional symbols in defining the metamaterial properties in the manuscript, when metamaterial bulk is usually defined with effective medium parameters (or equivalently band structure as mentioned if we are talking about refractive index only). **Following the suggestion, we have changed the r/i to the monopolar and dipolar polarizabilities (α_{00} or α_{11}), in Fig. 2 to 4, which are two common parameters together to represent single meta-atom property, e.g. Ref. [28] in using α to represent metamaterials made of one layer of atoms as well.** On the other hand, the connection to the bulk property (band structure/ effective medium parameters of multiple atoms) is indeed more

subtle. In our case of metamaterial atom in a 1D system, the near field in coupling the neighbouring unit cells (along the propagating direction) is not significant. The following simulation (Fig. S5) shows that the wavefront quickly goes to plane wave within 5cm, which is comparable to size of each atom (6.5cm), and much smaller than the wavelength of acoustic wave (34cm).

Figure S5 | The plane wave generation from meta-atom speakers Numerical simulations of meta-atom sources generating **a** monopolar (symmetric) and **b** dipolar (anti-symmetric) scattering fields. **c** Pressure field profiles measured at $y = 0$ (solid line) and $y = 0.02\text{m}$ (dashed line) for monopolar (blue) and dipolar (red) sources, where the wavefront goes to plane wave within 0.05m. Speakers modelled by $1.0 \times 1.5\text{cm}$ flat rectangular structure are 1.7cm away from each other and mounted in $2 \times 6\text{cm}$ rectangular waveguide.

Figure S6 | Response from multiple atoms **a** Effective medium parameters extracted from a single atom with convolution phase $\theta=180^\circ$. Solid lines (symbols) represent theoretical (experimental) results. Real (imaginary) part is shown in black (blue) color. **b** & **c** Transmission from cascading 2 or 3 atoms of the same configuration. Experimental results, for 2 atoms in **b** and 3 atoms in **c**, are shown in symbols while solid lines represent the theoretical results that are obtained from the previous single-atom property shown in **a**.

This is in the so-called transmission-line metamaterial regime (typical acoustic example: Fang et.al., Phys. Rev. Lett. 102, 194301 (2009) on acoustic negative refractive lens; typical electromagnetic example: Zhang et.al. Opt. Exp. 13, 4922 (2005) on fishnet metamaterial). **With this background, the single atom property also represents the bulk property when atoms are cascaded in the propagating direction.** Figure S6(a) shows the more traditional representation of effective medium parameters of the same type of atoms in Fig. 2 (with a smaller resonance strength and convolution phase 180°), now in terms of an anti-Lorentzian resonating reciprocal bulk modulus and a unit density (not shown here). Symbols / Solid lines

the experimental results/extracted theoretical model. Black/blue represents the real/imaginary part.

Then, by using these effective single-atom medium properties, we can calculate the expected two-atoms and three-atoms properties, transmission amplitude and phase spectra are shown in Fig. S6 b and c, as solid lines, which agree also to the experimental results shown in symbols, showing the validity in using single atom property in scaling up to the situation of a bulk.

We have added these new results in Supplementary information section C and we refer the readers to these new results in discussing the effective medium formula in main text now:

“We note that the near-field coupling (as there is no physical structures), if we periodically place identical atoms along the propagation direction, can be neglected, the effective medium parameters are still valid when we scale up the number of atoms (see Fig. S6 and the results about multiple unit cells in Supplementary Information section C)”.

Reviewer #3 (Remarks to the Author):

*This manuscript is concerned with the tunability of metamaterials in regard to the characteristic propagation of elastic waves. The authors identify several tuning methods from the literature (e.g., active material constituents) and argue that the time delay and range of response are hardly suitable for applications which require real-time adjustments. The authors propose a digitally constructed meta-atom -- consisting of an microphone-speaker assembly attached to a 1D waveguide -- for independently tuning the effective mass density and bulk modulus critical to the wave dynamics. Signals registered by the microphones are convoluted (in time) via a defined transformation and sent to the speakers as secondary monopole (linked to effective modulus) or dipole (linked to effective mass density) radiation. The theoretical/numerical results were experimentally confirmed. **The manuscript is clearly written, and presents original and potentially transformative concepts. Consequently, the review recommends the manuscript for publication.***

Reply: Thanks for the support on publication for our manuscript. We especially liked your summary on our work “*Signals registered by the microphones are convoluted (in time) via a defined transformation and sent to the speakers as secondary monopole (linked to effective modulus) or dipole (linked to effective mass density) radiation*”. **Thank you very much.**

Small edit suggestion: Equation 4 displays the imaginary unit "i" in both scripted and non-scripted form. One form should be used.

Reply: We have now unified to use the scripted “*i*” for the imaginary i , while non-scripted i used for indices. Similar modifications in main text are done as well.

Reviewers' Comments:

Reviewer #1:

Remarks to the Author:

The authors addressed adequately all the issues raised in my previous report. The manuscript presents an active metamaterial unit cell which can be tuned to have a monopolar and/or dipolar behavior, i.e. the building blocks of any acoustic response, without changing the unit cell physical structure and without being subjected to the usual restrictions of passive media (e.g. loss). The manuscript is timely and presents work of interest to the community. I therefore recommend its publication in Nature Communications.

Reviewer #2:

Remarks to the Author:

The authors have made satisfactory changes to all review comments.

Reviewer #1 (Remarks to the Author):

The authors addressed adequately all the issues raised in my previous report. The manuscript presents an active metamaterial unit cell which can be tuned to have a monopolar and/or dipolar behavior, i.e. the building blocks of any acoustic response, without changing the unit cell physical structure and without being subjected to the usual restrictions of passive media (e.g. loss). The manuscript is timely and presents work of interest to the community. I therefore recommend its publication in Nature Communications.

Reviewer #2 (Remarks to the Author):

The authors have made satisfactory changes to all review comments.

Reply:

Thanks for all the reviewers and editor for the support of publication of our manuscript. We also thank a lot on the constructive comments from both referees on improving our manuscript.